# Origin of the Chordate Notochord

Zihao Sui [1], Zhihan Zhao [2] and Bo Dong [2,3,*]

1 Department of Molecular Biology, University of Chicago, Chicago, IL 60637, USA; zsui@uchicago.edu
2 Sars-Fang Centre, MoE Key Laboratory of Marine Genetics and Breeding, College of Marine Life Sciences, Ocean University of China, Qingdao 266003, China; zhaozhihan@stu.ouc.edu.cn
3 Institute of Evolution & Marine Biodiversity, Ocean University of China, Qingdao 266003, China
* Correspondence: bodong@ouc.edu.cn

**Abstract:** The phylum of Chordata is defined based on the discovery of a coelom-like dorsal notochord in ascidian and amphioxus embryos. Chordata can be classified into three subphylums, Cephalochordata, Urochordata, and Vertebrata, united by the presence of a notochord at some point during development. The origin of the notochord, the signature anatomical structure of chordates, has been under debate since the publication of Alexander Kovalevsky's work in the mid-19th century that placed ascidians close to the vertebrates on the phylogenetic tree. During the late 20th century, the development of molecular and genetic tools in biology brought about a revival of studies on the evolutionary path of notochord development. Two main hypotheses for the origin of the notochord were proposed, the de novo theory and the axochord theory. The former states that notochord has developed de novo from the mid-dorsal archenteron of a chordate ancestor with simple morphology and no central nervous system nor notochord homolog. The putative notochord along the dorsal side of the animal is proposed to take on the signal functions later from the endoderm and ectoderm. An alternative hypothesis, the axochord theory, proposes that notochord has evolved from the mid-line muscle tissue, the so-called axochord, in annelids. Structural and molecular evidence point to the midline muscle of annelids as a distant homolog of the notochord. This hypothesis thus suggests a notochord-like structure in the urbilaterian ancestor, opposed to the consensus that notochord is a chordate-specific feature. In this review, we introduce the history of the formation of these views and summarize the current understandings of embryonic development, molecular profile, and gene regulatory networks of notochord and notochord-like structures.

**Keywords:** notochord; axochord; stomochord; ascidian; amphioxus; annelid; origin; evolution

## 1. Introduction

The notochord as an anatomical structure unites all chordates from lancelets to humans. It runs along the anterior–posterior (A–P) axis on the dorsal side of the animal and is surrounded by layers of notochord sheath. In vertebrates, the notochord is a precursor of the adult vertebrae [1]. The vacuolated notochord cells provide structural support to the embryo, serving as a turgid rod maintaining the shape of the tail and an anchor for muscle attachment to assist movement of the animal [2]. Arising from the dorsal organizer of chordates, the notochord also functions as a signaling center during development, releasing *Chordin*, *Noggin* and *Follistatin*, and the ventralizing ligand *Sonic hedgehog* (*Shh*) to adjacent tissues, participating in the formation of neural tubes and paraxial muscles [3,4].

The appearance of the notochord is an evolutionary landmark defining Chordata. The flexible and hard chord provides the basis for a more controlled and rapid movement and protection for the neural tube. The vertebrate notochord attracts sclerotome cells to migrate around itself and the neural tube, where they condense and differentiate to form the perinotochordal tube. This structure later develops into the vertebrate, and remaining notochord cells are relocated to the intervertebral regions, forming the nucleus pulposus of the intervertebral discs [5,6]. Failure of carrying this process to completion is associated

with the occurrence of chordomas, a rare cancer associated with notochord remnant that affects roughly one in a million people [1,7].

The evolutionary origin of the notochord is still debated to this very day. Could homologous structure be sought for in basal organisms? The enteropneust theory emerged as early as 1866, proposed that the notochord homolog could be found in hemichordate stomochord [8]. The proposed notochord homolog, hemichordate stomochord, is a short diverticulum projecting anteriorly from the buccal cavity, functioning as a fulcrum to facilitate burrowing for the marine invertebrate [9]. This view was supported based on the expression of *hedgehog* mRNA in hemichordate stomochord [10]. Studies challenged this view when modern molecular classification failed to find a common expression of key notochord transcription factors in stomochord [11–13]. Instead, a possible homology between hemichordate stomochord and pharynx-derived organ in chordates was proposed. The revived enteropneust theory of the 20th century onward, instead of trying to look for homologous structure in the hemichordate, has considered the notochord as a structure that developed de novo from the mid-dorsal side of the archenteron in chordates and took over signaling functions from ancestral endoderm and ectoderm [14]. Instead of a protostome–deuterostome ancestor already processing a prototype for notochord, this theory supports a simple ancestor without a centralized nervous system nor a notochord homolog. Around 1871, Alexander Kovalevsky, during his studies on embryogenesis of amphioxus and ascidians, found notochord structure in two invertebrates, amphioxus and ascidians. A few years later, the current definition of the phylum Chordata, including cephalochordates, urochordates, and vertebrates, was established by Haeckel [15]. Kovalevsky and colleagues then further proposed that a homolog for notochord might be found in the fibrous cells running along the nerve cord in annelids [16]. Later studies in the 19th century supporting the view of Kovalevsky proposed that the notochord shares an ancestral structure with the annelid ventral muscles [17,18]. In later supportive theories, annelid-like ancestors had undergone dorsoventral inversion during evolution into vertebrates. Hypotheses regarding how the transition had taken place varied. While one prominent hypothesis proposed that annelid muscles that migrated away from the nerve cord later transformed into cartilage [17], alternative hypotheses thought vertebrate notochords are homologs of fibers associated with the annelid nerve cord [18].

After long-term neglect, the annelid hypothesis received renewed attention in the new millennium. Following up on the studies connecting annelids to chordates in the 19th century, Arendt's group provided an investigation on the relationship between the annelid ventral muscle and notochord. In a study published in 2014, they named the hypothesized homologous structure in annelids as "axochord" based on structural and genetic evidence [19]. In response to this work, a subsequent review considering the axochord scenario pointed out two aspects missing from Arendt's study: identification of species bridging the wide evolutionary distance between annelids and chordates, and interaction network between the key transcription factors (TFs) expressed in the ventral mesodermal cells that specify and pattern the axochord [20]. A review paper in 2015 addressed these inquiries. In this review, the authors followed Remane's criteria for homology and Hennig's cladistic approach and expanded the search for possible axochord-like structures to a wide spectrum of invertebrate phyla, including Spiralia, Chaetognatha, and Ecdysozoa, which supported the evolution of notochord from an ancestral ventromedial muscle [21].

Kovalevsky, while focusing his studies on amphioxus in his early career, has once observed the unique developmental pattern that distinguished amphioxus as a group of deuterostomes possessing both vertebrates and invertebrate features [22]. Structural analysis in the 20th century has shown that the notochord of cephalochordates has muscular characteristics [23]. Modern molecular studies have also revealed that amphioxus, instead of ascidians, diverged earliest within Chordata, thus establishing tunicates instead of cephalochordates as the closest living relatives of vertebrates [24]. This places the amphioxus notochord as the potential link between the annelid axochord and the chordate notochord [25,26].

During the 19th century, a myriad of other hypotheses was proposed, which traced the origin to notochord to cnidarians, phoronids, and arthropods; however, these studies were mostly neglected in the late 20th century when interests in notochord evolution revived. Another hypothesis popular at the time, the nemertean theory stated that mesenchyme cells in nemertean probosci's coelom multiply to form a cartilaginous notochord [27]. This view had been partially refuted due to molecular phylogenies relegating nemertean to Lophotrochozoa [28]. In recent years, microscopy study has identified muscle formations on the ventral midline during the early stage of embryogenesis in the nemertean *Prosorhochmus* [29]. This ventromedian muscle is proposed to represent an axochord-like structure in nemerteans, which suggests the presence of axochord in the spiralian ancestor [21].

Molecular studies estimated that the divergence between chordates and ambulacraria had occurred before the Cambrian Era, around 570 MYA [30]. The cambrian fossil record provides a rich array of early deuterostome species, including hemichordate, cephalochordates, and primitive chordates, as well as extinct deuterostome phyla. The recent discovery of *Shenzianyuloma yunnanense*, a Cambrian vetulicolian possessing a notochord-like structure, provided potential novel insights into notochord origin [31]. Vetulicolians, a group of extinct deuterostomes characterized by their bivalved body plan; the anterior half of the body is a head-like structure containing gill-slits, and the posterior part of the body is segmented like annelids. This resemblance of both chordate and arthropod has caused phylogenetic challenges. Current consensus suggests that vetulicolians form a phylum among the stem-group chordates, though the exact position is unclear [32]. Anatomical analysis has tentatively placed the phylum basally close to the early Palaeozoic echinoderms, homalozoans [33]. Others suggest that the possible presence of an endostyle could place vetulicolians closer to chordates [34]. The discovery of a potential notochord in the fossil of *Shenzianyuloma yunnanense* complicated the problem. Depending on the phylogenetic position of the vetulicolians, this discovery in fossil could either mark a primitive notochord at the base of the deuterostomes, thus providing supports for a deuterostome ancestor with a notochord-like structure, or contribute to the classification of an archaic chordate-related phylum. Taking the bivalved body plan of vetulicolians into accounts, both possibilities would open up interesting questions regarding the origin of chordates and notochord.

This review will provide some considerations on the two competing hypotheses for the origin of the notochord, the de novo theory and the axochord theory, and evaluate the relationship between the muscular amphioxus notochord and the annelid axochord.

## 2. Embryonic Development of Notochord-like Structures

The phylum of Chordata includes three subphylums, Urochordata, Cephalochordata, and Vertebrata. Cephalochordates diverged from the ancestral lineage which gave rise to the other two subphyla, while urochordates and vertebrates form a sister group, known as the Olfactories [35]. The subphylums are united by a structure known as the notochord developed with dorsal mesoderm from the archenteron of the tail-trunk organizer of chordates. During this process, neural tubes and somites are induced adjacent to the notochord [36]. Whether the notochord possesses any homologous structures outside of the chordate phylum has been under debate. Based on the proposed similarities in structures and developmental processes, studies have sought notochord homologs in related bilaterian species such as annelids and hemichordates. Here, we provide a review of the developmental processes and structural characters of the notochord and notochord-like structures in different organisms from vertebrates to annelids (Figure 1).

In mice, the notochord tissue arises from the axial mesoderm on the midline of the embryo around 7.5 days after fertilization [37]. The vertebrate notochord forms in a convergent extension fashion similar to the development of notochord in ascidians, initiating from a 10-somite stage [38]. As precursors to vertebrae, the notochord plays a conserved role in attracting osteoblasts that originate in the sclerotome portion of the paraxial mesoderm to

form the vertebral bone [39,40]. Zebrafish notochord cells, representative of all vertebrate notochord cells, contain vacuoles that when lost lead to spine malformation [41]. The condensation of notochord cells into regions of the vertebral column where the discs are forming was proposed to be mediated by the notochord sheath in a squeezing fashion [42].

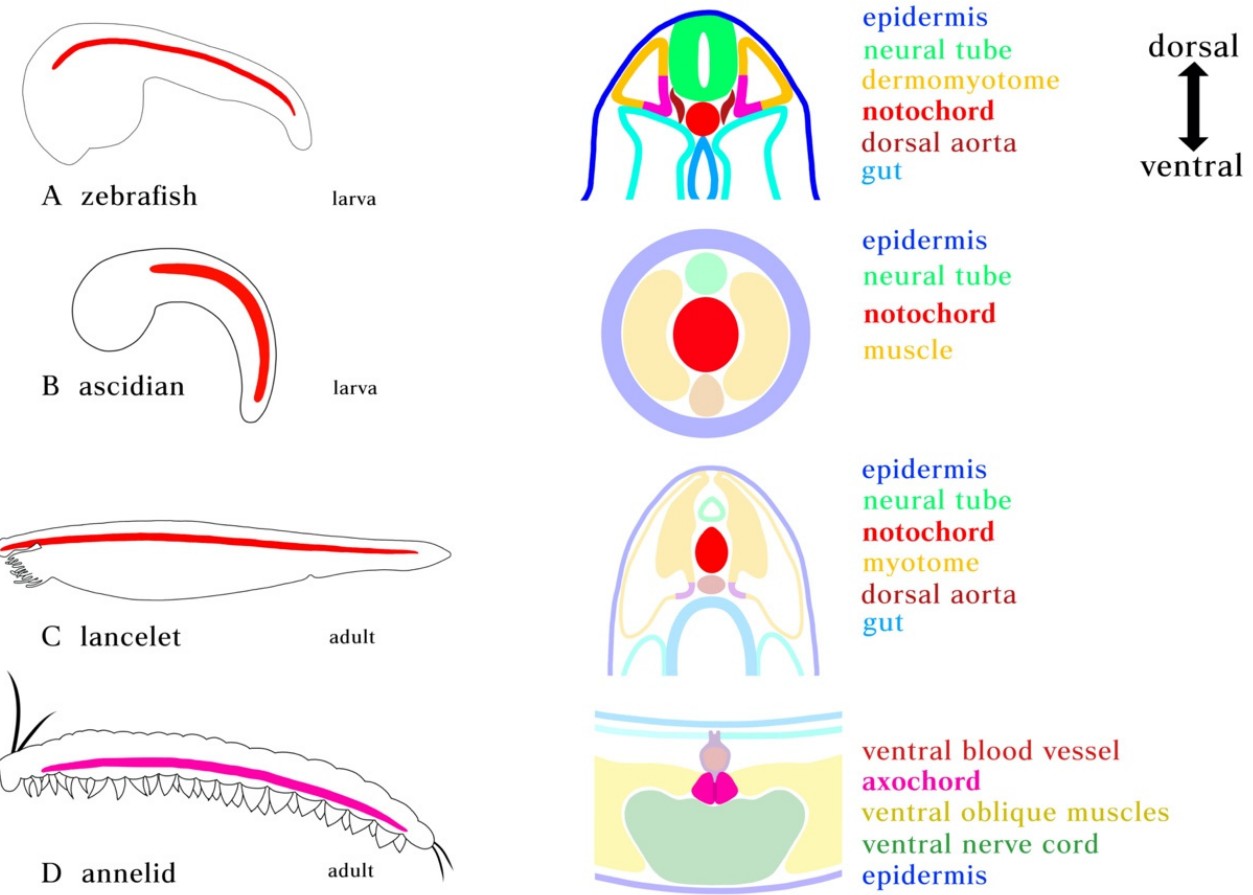

**Figure 1.** Position of notochord and axochord in bilaterians. (**A**) Zebrafish notochord. (**B**) Ascidian notochord. (**C**) Lancelet notochord. Notochord is positioned just ventral to the neural tube and dorsal to the gut, flanked by myotome. (**D**) Notochord homolog in annelid. Cross-section showing the position of the proposed axochord to the ventral mesentery, blood vessel, and nerve chord. Axochord is found to be dorsal to the nerve chord and ventral to gut of the animal. Red: notochord; Magenta: axochord; Green: nerve chord; Blue: epidermis; Yellow: mesoderm.

In ascidians, all morphogenetic processes of notochord take place in a group of 40 mitotically inactive cells that appear to be terminally differentiated. The notochord cells are derived from two separate lineages, a group of 32 anterior cells from the A7.3 and A7.7 cells in the blastomere, and eight posterior cells from the posterior blastomere B8.6. From the onset of gastrulation to neurulation, a semicircular arc around the anterior lip of the blastopore containing eight anterior notochord cell precursors and two posterior precursors undergoes two rounds of division to give the 40 cells [43,44]. Invagination and convergent extension change the structure of the notochord into a single column of disk-shaped cells [45]. Subsequent cellular elongation, mesenchymal–epithelial transition, and tubulogenesis lead to the formation of a full-length notochord with a continuous lumen going through it [46].

Notochord becomes defined around the same time in cephalochordates. Amphioxus gastrulation occurs by invagination. The gastrula consists of the outer ectoderm and inner mesendoderm, each one cell thick. At the late gastrula stage, the dorsal mesendoderm folds into three pleats in the anterior–posterior direction. The medial one develops into the

notochord while the two lateral ones form the anterior somite. These structures eventually pinch off from the archenteron pleats while retaining their relative positions. During the late neurula stage, the tissue around the blastopore starts to develop as the tailbud, giving rise to the remainder of the notochord [47]. Cephalochordate notochord contains central vacuoles flanked anteriorly and posteriorly by myofilaments [26]. In contrast, the vacuoles in ascidian *Ciona* spp. are placed between adjacent cells as a series of intercellular apical vacuoles during development [48].

Compared with chordates, hemichordates lack a definitive notochord lineage during development. In one of the proposed homologous structures of notochord, stomochord, cells are vacuolated and surrounded by a sheath, similar in tissue organization to a notochord [49]. It was also found to be a Hedgehog signaling center [11,12]. However, the appearance of hemichordate stomochord in the developmental process is delayed in comparison to the chordate notochord. Whereas ascidian and amphioxus notochord appears early in development, stomochord does not appear until the juvenile stage [11]. During the development of acorn worm juveniles, a stomochord anlage first appears as a short diverticulum protruding from the gut lumen anterodorsally into the protocol. This structure eventually becomes a rod-like protrusion from the dorsal buccal cavity [10,50].

In the proposed axochord theory, Arendt and colleagues attributed the axochord precursors to a group of mesodermal bands near the ventral side of the annelid *Platynereis* spp. larva at 34 h post-fertilization (hpf) by lineage tracing. Their study also revealed the movement of those mesodermal cells under neuroectoderm toward the midline until they contacted their bilateral counterparts, in a fashion reminiscent of the convergent-extension development of chordate notochord [19]. According to a comprehensive microscopy study on the development of the annelid *P. dumerilii*, the mesoblasts start to divide and produce the mesodermal bands near the vegetal pole around 7–13 hpf during gastrula and continues to grow in the hyposphere. Around 28 hpf, the dorsal longitudinal muscles appear in the larvae, while the ventral longitudinal muscles, the proposed homologous structure of the chordate notochord, become clearly visible slightly later, at around 32 hpf [51].

The dorsal–ventral axis in bilaterians is established by the opposing gradient of BMP and chordin [52]. The notochord of chordates and cephalochordates, as well as the stomochord of hemichordates, appear on the dorsal side of the embryo, while the annelid axochord is observed on the ventral side. This phenomenon is tentatively explained by the occurrence of an inversion event sometime during the evolution of chordate, which places the neural chords on the opposite sides of the embryo in vertebrates and insects. This proposed inversion of the dorsal–ventral axis, first hypothesized by Arendt and Nubler-Jung in 1994, occurred independently from the anterior–posterior axis and provides the alternative possibility of a complicated unilateral ancestor with a centralized nervous system [53]. Further modifications of the dorsal–ventral inversion theory suggest that the ancestral species do not need a centralized nervous system for the inversion, as in the case of hemichordate, which has a diffused nerve net [54,55]. In amphioxus embryos, BMP is normally expressed on the ventral side; overexpression of BMP causes ventralization of the embryo; under these conditions, *brachyury* expression in the notochord disappears [56]. Conversely, studies on body patterning of hemichordates have observed that mouth formation in hemichordates is repressed by BMP and the ventral tissue becomes dorsal-like. Chordates likely acquired a new mouth during evolution on their ventral side.

This theory explains the differences in the locations of the axochord and the notochord. Though the axochord and the notochord develop on opposite sides of the embryo, the dorsal–ventral inversion theory suggests that they could still be subjected to a similar genetic regulatory network. Hemichordates, which have not undergone the chordate specific dorsal–ventral inversion, should have their potential notochord homolog in the ventral region instead of the dorsal position, where the stomochord and pericardium resides.

### 3. Molecular Profiles and Cell Types of Notochord-Associated Tissues

Although the anatomical positions of the chordate notochords are similar, their composition and characteristics vary. Whereas mature ascidian notochord is a highly hydrolated and tubularized structure surrounded by a sheath composed of extracellular matrix (ECM), the vertebrate notochord is filled with cells with prominent central vacuoles. Amphioxus possesses a muscular notochord with vacuoles flanked by myofilaments. Annelid axochord is composed of muscle cells that lack observable vacuoles but is similar in its molecular characteristics with chordate notochord. Mature axochord is embedded in the fibrous sheath of the ventral nerve cord, serving as a point of attachment for transverse muscles [19] (Figure 2).

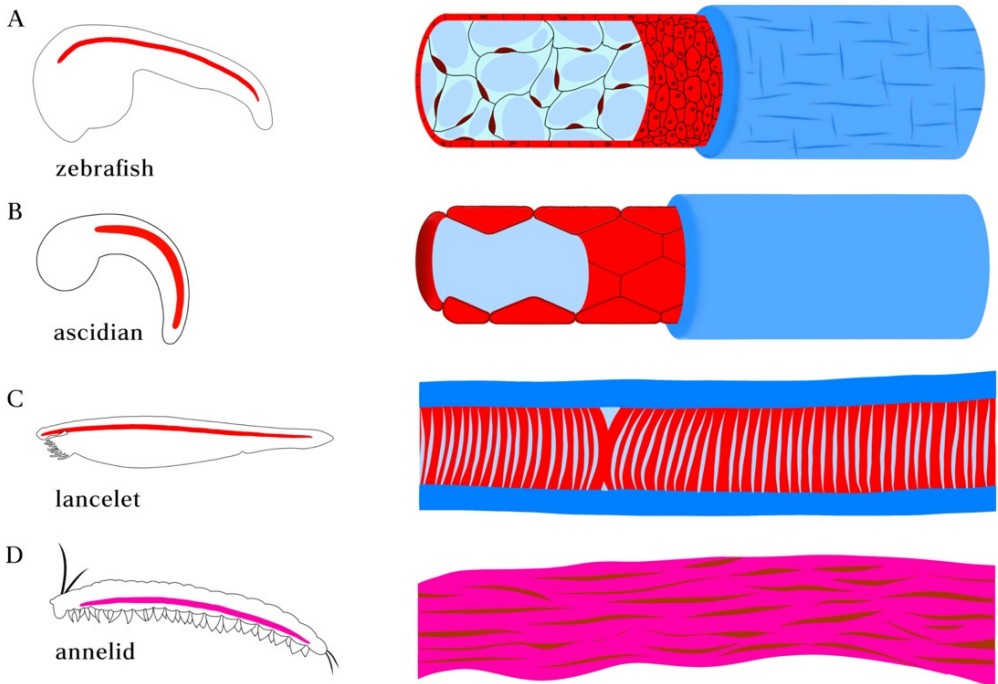

**Figure 2.** Morphologies of notochord in chordate species and axochord in annelid. (**A**) notochord in zebrafish embryo. The notochord in vertebrates is composed of a rod of highly vacuolated cells surrounded by epithelial (teleosts) or acellular (amniotes) sheath. (**B**) Ascidian (*Ciona*) notochord. The developing single-filed notochord cells move in polarized direction and undergo tubulogenesis, forming a long continuous fluid-filled lumen surrounded by sheath composed of basal lamina and a layer of circumferentially arranged extracellular matrix filaments. (**C**) Notochord in amphioxus lancelet. The muscular notochord cells contain vacuoles flanked by muscle fibers and surrounded by layers of laminin and collagen-based sheath. (**D**) Annelid axochord. The proposed homologous axochord is composed of a paired axial muscle spanning the ventral midline of annelid worms.

Notochord is the main axial structure of the vertebrate embryo. A number of mutations were found to cause notochord malformation in zebrafish, including the not homolog (*flh*), *brachyury* homolog (*ntl*), *foxa* and *tbx-c*. Additional mutations in *bozozok, gno, sleepy, tbx-c* were also found to affect notochord formation [57,58]. *Ntl* expression was found in the undifferentiated mesoderm and the notochord [59]. The homolog of not, *flh*, has shown specific and restricted expression in zebrafish embryos during development. In *flh* mutants, notochords are converted to mesodermal fate with an expression of *myoD*, and markers characteristic of paraxial mesoderm are found to be expressed in midline mesodermal cells [60]. In addition, the normal formation of zebrafish notochord was also found to be dependent on the specialization of early chordamesoderm by Nodal and EGF-CFC family co-receptors [61,62]. The notochord sheath is composed of the basal lamina and circumferentially arranged extracellular matrix filaments [63].

The amphioxus notochord was observed to possess muscle-like property and is composed of mixed cell types [23]. The center of the notochord is occupied by contractile cells containing myofibrils and expresses muscle-related genes, while the peripheral cells are non-contractile and express collagen genes [64,65]. Early ontogeny of the notochord of *Branchiostoma lanceolatum* based on electron microscope showed that notochordal myofilaments are observed to be capable of contraction [26]. EST analysis showed that of the 128 cDNA clones from amphioxus notochord cells showing similarity to known proteins, 22% of them are muscle genes. Moreover, it is observed that amphioxus notochord expresses both components of the actin-linked regulation and components of myosin-linked regulation, as well as intracellular hemoglobin. This suggests that the muscle-contractile components of the amphioxus notochord represent a mixture of vertebrate-type striated and smooth muscle components [66,67]. The uniquely expressed actin in the amphioxus notochord is neither of cytoplasmic type nor of muscle type and was named the amphioxus notochord actin [66].

Ascidian notochord develops from a single-filed row of cells arising from two separate lineages. The primary lineage contains 32 cells and the posterior secondary lineage of the notochord is composed of 8 cells. Those two components of the notochord show a distinct developmental pattern as well as a variable gene expression profile [68]. While some genes including *brachyury* and *tropomyosin-like* are expressed uniformly along the notochord, differential expressions were found in *TGF-beta*, *ERM*, *Bcam*, *TGF-β*, etc. [69,70]. In addition, ascidian notochord is deprived of *hox* expression, which marks the notochord of other chordate groups and the proposed annelid axochord, likely due to clade-specific loss of expression during evolution after the split between Urochordata and Vertebrata [21].

Hemichordate stomochord has failed to be identified as an expression site for the key notochord genetic markers [11]. Of the key regulators of notochords, only *hh* is found in hemichordate stomochord, while *hh* is also expressed throughout the endoderm [9,71]. The key transcription factor, *brachyury*, is found to be expressed in the archenteron invagination region and stomodeum invagination region of gastrulae instead of stomochord [72]. FoxE, a transcription factor expressed in the pharynx of chordates, is found to be specifically expressed in the stomochord. Interestingly, the key notochord transcription factors appear to be dispersed in their expression domain in acorn worms, suggesting that maybe there lacks a homologous structure to notochord in hemichordate [11].

Seven TFs (*brachyury*, *foxA*, *foxD*, t*wist*, *not*, *soxD*, and *soxE*) and eight effector genes (*colA1*, *colA2*, *chordin*, *noggin*, *netrin*, *slit*, and *hedgehog*) which in combination uniquely define notochord fate in chordates were identified by Arendt's group [21]. Of those seven TFs, only *not* is absent from the ventral midline muscle of *P. dumerilii* [19–21]. Compared to the expression profile of notochord genes in different chordate species, the axochord cells express myosin heavy chain 1–4, which is absent from vertebrates and urochordates but present in the muscular amphioxus notochord [19]. This observation is in agreement with the suggestion that the mesodermal characteristic of amphioxus notochord is a relic of early homologous structures and the fact that cephalochordate has diverged first from the other two subphylums during evolution [24].

## 4. Molecular Components Belonging to the Notochord Gene Regulatory Networks (GRNs) in Ascidians, Amphioxus, and Annelids

A complex network of genetic regulation induces and maintains notochord fate during development. While the specific downstream pathways in the formation of notochord or notochord-like structure remain to be elucidated, the fundamental process of axial structure formation in bilaterians is widely conserved.

Here, we choose to examine four essential transcription factors including beta-catenin, foxa, *brachyury*, and *tbx2/3* from the perspective of their expression patterns during the development of the ascidian notochord. All these genes are versatile TFs that direct a myriad of developmental processes, whether their interactions in amphioxus and more basal species resembles the serial regulation in ascidian remains to be investigated. The downstream TFs act to reinforce the morphogenic effect of upstream activator, triggering

the expression of notochord genes, and form an interplaying and regulated cascade leading to notochord formation (Figure 3). More detailed GRNs could be found in previously published reviews [73,74].

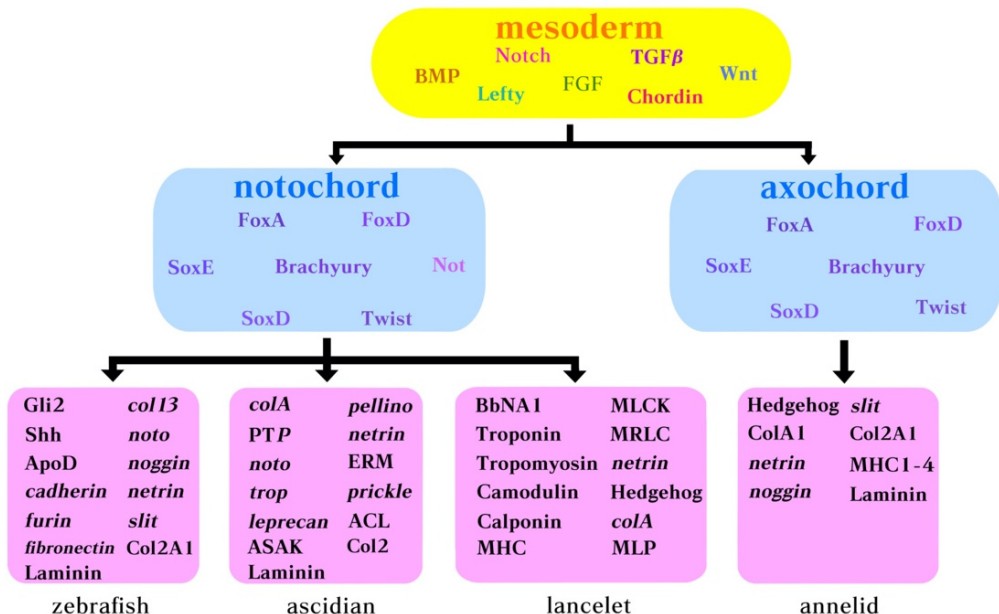

**Figure 3.** Molecular components belonging to the notochord GNR in ascidians, amphioxus and annelids. Yellow box: body morphogens acting early during development. Pink box: genes expressed by the notochord in zebrafish, ascidian, lancelet (amphioxus), and annelid.

### 4.1. Wnt/Beta-Catenin

The Wnt/beta-catenin pathway plays a fundamental role in the A/P axis establishment and is found to function in a conserved manner in a myriad of species [75]. Beta-catenin is found in all metazoans and possesses dual roles in structure and signaling. At cellular junctions, it interacts with E-cadherin and alpha-catenin to regulate the actin skeleton and maintain cellular structure. In the cytosol, free beta-catenin is normally phosphorylated and degraded by a destruction complex in the absence of Wnt signals. Introduction of Wnt ligands to transmembrane receptors triggers molecular events that eventually lead to the inhibition of the destruction complex and enables beta-catenin to be translocated into the nucleus. The nuclearization of beta-catenin enables the formation of a complex with other transcription factors and the activation or repression of target genes. Beta-catenin signals have pivotal roles in the development of axial structures.

Overexpression of beta-catenin was observed to induce a second A/P axis in *Xenopus* embryos, while it was also observed to cause exclusion of cells from notochord fate and contribute to the adoption of somite fate [76,77]. Depletion of beta-catenin in *Xenopus* causes shortening of the A/P axis and the ventralization of dorsal tissues [78]. Murine beta-catenin null mutants fail to develop a normal A/P axis and show specific inhibition of dorsal mesoderm formation [79,80]. In amphioxus, enhanced cWnt activity and beta-catenin nuclearization concur with an expanded expression of *brachyury*, a notochord marker gene, and the inhibition of dorsal mesoderm specification [81]. Arendt and colleagues' study also showed that over-expression of beta-catenin in the axochord converts the axochord into lateral muscle fate in the annelid *Platynereis* [19]. It was observed that suppression of the Wnt/beta-catenin pathway caused a decrease in *brachyury* expression in the notochord of ascidian *Ciona* [73,82]. In ascidians, maternal beta-catenin nuclear localization activates *foxa* expression. FoxA, along with other transactivators, in turn, activates the expression *of brachyury and tbx2/3bx2/3 in* the developing embryo [83,84].

### 4.2. FoxA

During the development of the model organism *Ciona intestinalis*, the forkhead transcription factors, FoxA and FoxD, together with maternal p53 activate zinc-finger gene *zincL/N*, which then binds and contributes to the activation of the T-box gene *brachyury* essential for notochord formation [85–87]. In addition, FoxA was observed to act synergically with Brachyury to activate downstream targets [88].

Simultaneous loss of function of FoxA family TFs causes a complete absence of all axial structures including the notochord in zebrafish embryos; instead, the former axial tissues are reconfigured to develop into paraxial mesoderm and neuroectoderm, while overexpression of *foxa2* and *foxa3* enlarges axial mesoderm at gastrula stage. Similar phenotypes were observed in vertebrate *Xenopus* [89].

FoxA-a and ZincL function as early determinants of the 16-cell blastomere patterning in ascidians. FoxA-a is expressed in the A-line progenitors at the 32-cell stage and persists till the 64-cell stage, which gives rise to the anterior 32 notochord cells. FoxA is found across tissue types including neural, mesenchyme, and notochord cells [83].

Amphioxus paralog of FoxA, FoxAa, were detected in the axial dorsal mesendoderm corresponding to the presumptive notochord territory, as well as in mesendoderm cells of the archenteron floor. At the late neurula stage, its expression in the notochord was restricted to the most anterior and posterior tips of the embryo, while the endodermal expression was restricted to the middle region of the gut. At the larva stage, the expression is found at the anterior tip of the notochord, the tailbud, and in the gut [90]. Both amphioxus and ascidian *foxa* have expression domains in the endostyle, which was proposed to be related to thyroid evolution [91].

FoxA could be found in the vegetal plate of the hemichordate blastula, giving rise to the endomesoderm. Throughout development, expression persists in the endodermal region but is excluded dorsolaterally from the regions that give rise to the gill pores. From 48 hpf on, expression is also found in the anterior ectoderm in the collar groove region. In juveniles, this circumferential expression marks the most anterior collar region while being excluded from the dorsal midline [92].

The expression of *foxa* in *P. dumerilii* was studied in detail with in situ hybridization. The expression starts at the cleavage/early gastrulation stage (6.5 hpf) and was subsequently observed in the region of the ventral and the lateral lips of the blastopore. Expression in mesoderm is also observed in the late cleavage stage. Past gastrulation, its expression becomes centered around the midline structures including precursors of foregut anlage and the stomodeum. Expression is also found in the invaginating stomdeal, the midgut, and brain in larvae and juvenile states [93].

### 4.3. Brachyury

Brachyury was recognized as the most important transcription factor in notochord formation. Ectopic expression of *brachyury* causes trans-differentiation of endoderm into notochord cells [73,94]. Overexpression of distant *brachyury* orthologs from annelid *P. dumerilii* and the cnidarian *H. magnipapillata* could induce mesodermal fate in *Xenopus* animal caps, implying a highly conserved mesoderm patterning character across species [95]. Induction of notochord fate is observed in ascidian embryos misexpressing *brachyury* from cephalochordates, hemichordates and echinoderms [96]. The original function of Brachyury is associated with morphogenesis instead of cell differentiation. Expressed in the blastopore region across metazoan species, it facilitates invagination of the endomesodermal germ layer. Secondary expression domain and function are acquired sometime during chordate evolution, where *brachyury* becomes expressed in the notochord forming region of the dorsal midline [56,73]. The expression of *brachyury* in ascidians is controlled upstream by a series of transcription factors and morphogens. Input from beta-catenin acting through FoxA, FGF-MAPK pathway, maternal p53 and ZicL/N in combination defines the expression domain of *brachyury* [87]. Downstream targets of Brachyury include a number of genes essential for notochord development, *prickle*, *ERM*, *noto3*, and *noto4* [87].

Brachyury has a conserved role in mesoderm differentiation in vertebrates: it is first expressed in the marginal zone of the late blastula, the presumptive mesoderm, but becomes restricted to the notochord and tailbud during gastrulation (i.e., axial and posterior mesoderm) [97]. In experiments performed with zebrafish, cells of the notochord domain form mesenchyme in *ntl* (*brachyury*) mutants [60]. Ectopic expression of *brachyury* in *Xenopus* during early embryo development could convert endoderm cells to mesodermal fate [98,99].

The primary expression of *brachyury* was lost and the expression is restricted to the notochord in ascidians [95,97]. *Branchiostoma*, on the other hand, shows a broader expression domain of brachyury that includes the muscular notochord and the somite flanking it. Although experiments have argued that ectopic misexpression of Ci-Bra transforms endoderm cells into notochord, recent genetic analysis brought new perspectives [100]. The study has demonstrated that misexpression of *brachyury* under *foxa* promoter induces many target genes, but the majority of notochord-enriched genes show no statistically significant response to ectopic *brachyury* expression. The limited overlap between ectopically induced target genes, notochord enriched genes, and those genes previously identified as putative targets of Brachyury implies the presence of other unidentified important regulatory elements contributing to notochord formation [101].

Brachyury is expressed in two separate locations during the embryogenesis of hemichordates. The first *brachyury* signal restricted to the blastopore of the base of the invaginating archenteron is detected at about 18 hpf. During the next phase of hemichordate gastrulation, the blastopore closes and then opens again. The signal remains at the base of the archenteron during gastrulation until it disappears in the 10-day-old larvae. Another *brachyury* expression domain becomes evident as early as the middle gastrula stage in the region that eventually forms the mouth or stomodeum. The signal in the stomodeum remains during the formation of the mouth and disappears prior to the disappearance of the archenteron signal [72].

Brachyury expression is first detected at 8 h of development in vegetal cells around the closing blastopore during embryogenesis of *P. dumerilii*. The blastopore closes in the middle and becomes slit-like, leaving two openings at either end, which will form the foregut and hindgut, respectively [97]. Post-gastrula, *Pd-bra* demarcates the outline of the closed, slit-like blastopore in the ventromedial region and is found in the developing ventral foregut (stomodaeum), ventral midline cells, and developing hindgut (proctodaeum). Brachyury is not detected in midline cells of 36 h or 48 h larvae [102]. Expression reappears along the midline of the developing young worm around 72 hpf in parts of the cellularizing midgut, facing the developing nerve cord [103].

### 4.4. Tbx2/3

An important downstream target gene of *brachyury* involved in notochord formation is the T-box transcription factor *Tbx2/3*. Tbx gene is present in all the major metazoan lineages examined and diverges into Tbx2 and Tbx3 at the root of vertebrates [104]. It was shown to be able to act as both transcriptional activator and repressor [105]. *Tbx2/3* is involved in a wide range of developmental processes in chordates, including ECM modeling, cell death, cell migration, and notochord developments, where it regulates/maintains the target genes of Brachyury [84].

The most conserved expression in vertebrates of Tbx2 and Tbx3 is found in neural crest cells forming the sensory dorsal root ganglia [106,107]. Tbx-c, zebrafish homolog of *tbx2/3*, was found to act downstream of *ntl* (*brachyury* homolog) and play a role in the specification of late notochordal precursor cells and the formation of the differentiated notochord. At the end of gastrulation, *tbx-c* expression is observed in the prospective ventral forebrain, the single eye field and the presumptive notochord. However, after 12 hpf, the expression of *tbx-c* in the midline is restricted to the posterior end of the notochord. Overexpression of *tbx-c* causes expansion of the notochord, whilst *tbx-c* mutants show attenuation of notochord formation [58]. The expression pattern in *Xenopus* similarly spans multiple tissue types.

*Xltbx2* was found to have an expression in the neural crest cells and *Xltbx3* in the ventral spinal cord [106].

*Tbx2/3* is activated during the neural plate stage of development and expressed in the notochord, CNS, and epidermis cells in the head area in ascidians. The expression in notochord is dependent on *brachyury* expression. *Tbx2/3* acts downstream of Brachyury in notochord formation to reinforce the effect of Brachyury and, in some cases, limit the notochord activity of brachyury targets. Defection in *tbx2/3* expression has impaired the convergent extension mechanism of notochord formation [84].

*Tbx2/3* in amphioxus is first detected around the lip of the blastopore during early gastrulation, marking invaginating mesendoderm cells. *AmphiTbx2/3* is expressed in the dorsal neural tube, regions of ventral gut endoderm, and some surface ectoderm cells. Diffuse expression is also observed in a subset of notochord cells along the entire rostrocaudal axis. Signal persists in the posterior mesendoderm, dorsal neural tube, gut endoderm, and surface ectoderm during neurulation and becomes diminished around the 4-day larva stage [104].

*Tbx2/3* has a defined expression domain along the dorsal margin of the embryo on the BMP side despite the lack of a definitive notochord homolog in hemichordate. In the hemichordate *S. kowalevskii*, *tbx2/3* expression first begins during gastrulation in a sector of the ectoderm. The expression domain then concentrates into a dorsal stripe, with the strongest expression shown posterior to the telotroch ciliated band of the animal. Anteriorly, the expression of *tbx2/3* extends ventrally and forms a circle near the first-gill slit [55].

*Tbx2/3* was similarly found on the dorsal part of the embryo through development in ectodermal, endodermal, and mesodermal precursors of polychaetes. Expression is initially detected in a few blastomeres that occupy the dorsal midline and remain centered around this area through the developmental process. Expression declines in vegetal areas during gastrulation, but dorsal expression continues during gastrulation in all three germ-layer precursors [103]. In general, the expression pattern of *tbx2/3* is associated with dorsal organ specialization, with a secondary role in notochord establishment. Whether this key transcription factor is expressed in the axochord of annelid *P. dumerilii* remains to be investigated.

Overall, the three transcription factors share expressions along the midline region in tunicates, amphioxus, and the polychaetes, while the specific expression domain of *tbx2/3* in relationship to the ventral midline muscle remains to be elucidated. In hemichordates, which share a closer phylogenetic relationship with chordates yet possess a simple body plan, these notochord marker genes appear to be dispersed through the body with little resemblances to chordate species, a phenomenon that could perhaps be attributed to its close relationship with echinoderms.

## 5. Conclusions

The notochord has traditionally been viewed as a structure uniquely defining the phylum Chordata. The various attempts made by 19th-century scientists proposed possible homologs of chordate notochord but did not reach a definite conclusion regarding the origin of this essential structure. The revival of interests in notochord's origin story brought forward two alternative views: the axochord origin theory, which focuses on the notochord-like structure in *P. dumerilii*, and the notochord novelty theory, which maintains the view of a notochord unique of the chordates. The notion of a notochord ortholog in distant protostome species would imply the presence of a homologous structure in the last common ancestor of deuterostomes and protostomes. Though the molecular profile defining notochord structure was identified by Arendt's group, the complex regulatory network connecting the TFs to their downstream effector genes has yet to be elucidated. In the closely related non-chordate species, hemichordate, Hh signaling that normally marks the notochord is found in the stomochord, which lacks other notochord markers. Brachyury and FoxA, two key TFs of notochord formation, are also found in the stomodeum

invagination of polychaetes and hemichordate. The partially overlapping characteristic of stomodeum and notochord in basal species opens up the possibility of the presence of an ancestral structure uniting the two. On the other hand, this review intends to point out that Kovalevsky's hypothesis, which states that amphioxus might be the intermediate stage from the development of non-chordates to vertebrates, is a direction worthy of future investigation. Cephalochordate diverges first among the three subphylums of Chordata and was viewed as a less derived clade among the three. Considering the muscular properties of amphioxus notochord and the phylogenetic position of amphioxus in light of the annelid axochord hypothesis offers a new perspective on the origin and evolution of notochord. Studies in the new millennium have paid little attention to the possible homology between muscular amphioxus notochord and presumptive axochord of annelids. Molecular studies comparing the gene expression profiles of the two structures, such as examination of the possible similarities between axochord actin and the unique amphioxus notochord actin, would potentially provide further information on the origin story of the notochord. The hypothesis regarding the possible links between the annelid ventral muscle and the chordate notochord poses a series of new questions. If the homology between notochord and structures in non-chordates species is proved to be plausible, then how did the notochord evolve from its primitive forms to the layered and vacuolated form seen in chordates? What drives the appearance of sheaths and internal vacuoles in chordate notochord? While considering structural evolution, investigations on the underlying process of the notochord acquisition of signaling center and dorsal organizer functions would also be worthy of future studies.

**Author Contributions:** Conceptualization, B.D. and Z.S.; writing—original draft preparation, Z.S., B.D., Z.Z.; writing—review and editing, B.D.; visualization, Z.Z.; supervision, B.D.; project administration, B.D.; funding acquisition, B.D. All authors have read and agreed to the published version of the manuscript.

**Funding:** This research was funded by the National Key Research and Development Program of China (Grant No. 2019YFE0190900) and the National Natural Science Foundation of China (Grant No. 31771649).

**Informed Consent Statement:** Not applicable.

**Conflicts of Interest:** The authors declare no conflict of interest.

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
