# Peer review of "Origin of the Chordate Notochord"

_diversity, doi:10.3390/d13100462_

Round 1

Reviewer 1 Report

The authors of this manuscript propose a link between the chordate notochord and the annelid/arthropod axochord. This is an interesting hypothesis, and might even find some support in the Cambrian fossil record, if we consider the presence of both arthropod-like and chordate characteristics in the enigmatic vetulicolians (Geosciences 20199(8), 354; https://doi.org/10.3390/geosciences9080354). The authors provide an interesting review of the relevant genomics and developmental biology, but they must take this a step further and provide an expanded discussion of the Cambrian fossil record. Also, English language usage in this manuscript requires extensive revision.

Comments on the manuscript:

abstract has major grammar issues

line 93 'Ecdysozoa' not 'Ecedysozoa'

lines 100-101 poor grammar here

line 247 period after etc

line 337 'in detail' should replace 'in details'

line 374 'analysis' should replace 'an analysis'

line 390 revise to read 'remains at the base'

line 484 'polychaetes' should replace 'polychaetas'

line 498 revise to read 'of the notochord'

Author Response

Thank you for your comments and suggestions. We have added a part on the vetulicolians and Cambrian fossil record in the introduction (line 138-158), and the abstract has been rewritten. Several other grammarical revisions are made, and the manuscript was proofread by a native speaker.

The page numbers in parenthesis refer to the original page number in the manuscript before edits.

Line 115 (93) 'Ecedysozoa' from 'Ecdysozoa'

Line 126-127 (100-101): sentence revised to read “During the 19th century, a myriad of other hypotheses have been proposed, which traced the origin to notochord to cnidarians, phoronid, and arthropods;”

Line 296 (247) period added after etc.

Line 390 (337) 'in details' changed to 'in detail'

Line 427 (374) 'an analysis' changed to 'analysis'

Line 443 (390) revise to 'remains at the base'

Line 499,508,537(484) 'polychaetas' changed to 'polychaetes'

Line 558 (498) revise to read 'of the notochord'

Reviewer 2 Report

This is a nice summary of hypotheses on the origin of the notochord, focusing on the two hypotheses that have been the most widely discussed in the recent literature: the novelty hypothesis and the axochord hypothesis. The presentation and discussion of data are objective and mostly solid, and all main references are cited.

I noticed a few inaccuracies that should be corrected:

  • the phrase "lower animals" is often used - this is outdated terminology. Instead, use "invertebrates" (or "non-chordates") when applicable.
  • l. 33: "The notochord as an embryonic structure unites all chordates..." The notochord is also an adult structure, eg in amphioxus, appendicularian tunicates, and even some vertebrates such as the coelacanth.
  • l. 45: "Notochord ossifies and is subsequently replaced by the skeletal tissues, forming the vertebrate in higher organisms" (here, "vertebrate" should be "vertebrae") This is not what happens. Vertebral skeletal tissue comes from the scleretome of the somites, and ultimately replaces the notochord. The notochord itself degenerates and only persists between vertebrae, in the form of spinal discs.
  • l. 57, the enteropneust stomochord is said to be "functioning as a fulcrum to facilitate undulatory swimming" - enteropneusts almost never do undulatory swimming. The stomochord assists in burrowing.
  • l. 102 "phoroids and anthropods" -> phoronids and arthropods
  • l. 119 "Olfactories" -> Olfactores
  • l. 130 "around 7.5 days after fertilization" - in which species?
  • l. 134 "Zebrafish notochord cells contain vacuole" all vertebrate notochords have vacuoles
  • l. 492: "Amphioxus" This should not be italicized or capitalized as it is not a Linnean name (the Linnean name of amphioxus is Branchiostoma)

Author Response

Line 91, 542 ‘lower animals’ changed to ‘invertebrates’ and ‘non-chordates’

Line 52 (33) ‘embryonic‘ changed to ‘anatomical’.        

Line 65-69 (45) changed to read ‘The vertebrate notochord attracts sclerotome cell to migrate around itself and the neural tube, where they condense and differentiate to form the perinotochordal tube. This structure later develops into the vertebrate, and remaining notochord cells are relocated to the intervertebral regions, forming nuclei pulposi. ’ Thank you for pointing the inaccuracy.

Line 78 (57) ‘swimming’ changed to ‘burrowing’

Line 127 (102) changed to read ‘phoronids, and arthropods’

Line 167 (119) Olfactories changed to ‘Olfactores’

Line 178 (130) sentence revised to read ‘In mice, the notochord tissue arises from the axial mesoderm on the midline of the embryo around 7.5 days after fertilization’

Line 182-184 (134) sentence revised to read “Zebrafish notochord cells, representative of all vertebrate notochord cells, contain vacuole that when lost lead to spine malformation”

In the Conclusion section, all incidents of ‘Amphioxus’ has been changed to ‘amphixous’.

Reviewer 3 Report

The work reads generally ok but lacks novelty both in reporting of new datasets and hypothesis or criticism that would contribute to further discussion in the field. The work does not add in fact compared to previous reviews. For instance, the works by Annona et al,. 2015 and Brunet et al., 2015 include and extend all the points of discussion (including developmental origin, position, structure, gene expression and the hemichordates debate) which are only touched here. No clear conclusions emerge from the section of the gene regulatory network, which does not provide sufficient new hints as compared to previous considerations on the topic. Also, the review promises to focus on Amphioxus as a possible model system for future comparison given the possible link it represents between the annelid axochord theory and the origin of a bona fide notochord. Nevertheless, the authors fail to engage in an in-depth discussion on this or provide new insights in this direction. Besides identifying the two original hypotheses the work neither propose scenarios nor acknowledge those proposed by others for the formation of a complex vacuolated notochord

In details, issues that need to be fixed are:

Line 23

the authors are not accurate when they report the hypothesis tested by Arendt and colleagues. Indeed, the “axochord” hypothesis does not suggest the “presence of an ancestral notochord”. It actually speculates on the possibility that the notochord originated from a series of events which modified an ancestral ventromedial muscle which predated the origin of chordates. Therefore, the authors should more carefully report the hypothesis tested in the literature.

Line 50, Line 70 and elsewhere

“Lower organisms” is not entirely correct. An alternative and more scientifically correct term such as “Basal”should be used.

Line 72

“is established” should be replaced with “was established” and authors should not refer to “Alexander” but rather to “Alexander Kovalevsky”or simply to “Kovalevsky”.

Line 85-92

“Analogous” should be instead “homologous”. Also, the authors of the 2014 (Lauri et al.,) and later 2015 ( Brunet et al.) work followed a comparison based on Remane’s criteria for assessing experimentally evolutionary homology. These are specific quality (1), position (2), continuity (3). The authors should mention here the use of these fundamental criteria for comparison.

Line 212 and 286

The title of the figure is misleading. The axochord is proposed as homologous to the notochord and not as an annelid notochord. The title should be something like “position of notochord and axochord in bilaterians”or similar.

In fig.1 and 2 the authors should specify the relative position of the blood vessels in the different taxa representatives.

Line 384

It is not clear what the authors want to discuss in evolutionary terms when they pair the regulation of Brachyury at the bp by modulation of Wnt/beta-catenin and the ability of a fine tuning of beta-catenin, whose nuclear activity confers the paraxial cell types identity in both vertebrates and annelids . Those two events might be neither developmental nor evolutionary linked and the authors should clarify what their comparison is and which conclusions can result form this.

I find Figure 3 very confusing. A clear gene regulatory network is not depicted but I see rather boxes with a number of genes and a distinction between the notochord-axochord specifics is not opportunely shown. Zebrafish notochord express Col2A1 and this should be included as well as Laminin. Repeating for three times (three figures) the same animal body plans (same illustrations) should be avoided.

Equally confusing is the paragraph on the gene regulatory network. What are the comparison the authors want to make and which considerations can be learn from this? Authors should restructure better this paragraph and the figure.

Round 2

Reviewer 3 Report

The authors have improved the manuscript following the majority of the reviewers' comments. I still see a lack of novelty and of detail in the examination of the different hypotheses and on the 1. embryonic development, 2. molecular profile and 3. GRN. It would be beneficial to focus better on few key points for each of these topics, reducing the confusion and also improve figure 3 accordingly.  

Some comments here: 

Line 134

“Inquisitions” is not an appropriate term in this context

Line 136

“Chordate” should be “Chordates”

Line 134-136

Here, the authors did not “use” those criteria “to support”. Indeed, I would rather rephrase as such:

“ in this review the authors followed Remane’s criteria for homology and Henning’s cladistic approach, and expandied  the search for possible axochord-like structures to a wide spectrum of invertebrate phyla, including Spi ralia, Chaetognatha, and Ecdysozoa, which supported the evolution of notochord from an ancestral ventromedial muscle”.

Line 263 and elsewhere

“vacuole” should be “vacuoles”

Line 468

The title of the paragraph does not make sense. The authors should rephrase in something like: “ Molecular components belonging to the notochord GNR in ascidians, amphioxus and annelids”

Line 470-472

“suppress” is repeated twice and the whole sentence is not clear. Ciona belongs to ascidians. Do the authors refer to other ascidians besides Ciona? A reference work is not provided! Moreover, the different evidence for the role of beta-catenin should be better presented in the different taxa. For instance, in amphioxus it is about dorsal organizer and not cell type sorting and differentiation between somite and notochord.

Figure 3 and paragraph on GRN 

I do not find the figure much improved. The discussion on the GRN remains confusing as the figure, at least the author should rearrange the figure in such a way that a clear distinction between patterning factors and differentiating TF factors would emerge, possibly within a GRN between the genes wherever is known at least for vertebrates and others.. along the lines of what published by Di Gregorio, 2020 https://doi.org/10.1016/bs.ctdb.2020.01.002.

Despite my comment, in the revised figures the authors failed to mention both Col2A1 in zebrafish notochord or Laminin, which is present in Zebrafish, Ciona and annelids. This should be mentioned.

Author Response

The authors have improved the manuscript following the majority of the reviewers' comments. I still see a lack of novelty and of detail in the examination of the different hypotheses and on the 1. embryonic development, 2. molecular profile and 3. GRN. It would be beneficial to focus better on few key points for each of these topics, reducing the confusion and also improve figure 3 accordingly.  

Some comments here: 

Line 134

“Inquisitions” is not an appropriate term in this context

Line 136

“Chordate” should be “Chordates”

Line 134-136

Here, the authors did not “use” those criteria “to support”. Indeed, I would rather rephrase as such:

“ in this review the authors followed Remane’s criteria for homology and Henning’s cladistic approach, and expandied  the search for possible axochord-like structures to a wide spectrum of invertebrate phyla, including Spi ralia, Chaetognatha, and Ecdysozoa, which supported the evolution of notochord from an ancestral ventromedial muscle”.

Line 263 and elsewhere

“vacuole” should be “vacuoles”

Line 468

The title of the paragraph does not make sense. The authors should rephrase in something like: “ Molecular components belonging to the notochord GNR in ascidians, amphioxus and annelids”

Line 470-472

“suppress” is repeated twice and the whole sentence is not clear. Ciona belongs to ascidians. Do the authors refer to other ascidians besides Ciona? A reference work is not provided! Moreover, the different evidence for the role of beta-catenin should be better presented in the different taxa. For instance, in amphioxus it is about dorsal organizer and not cell type sorting and differentiation between somite and notochord.

 Response:

Thank you for your comments and suggestions. The following readjustments have been made:

Line 134: “inquisitions” changed to “inquiries”

Line 136: “chordate” changed to “chordates”

Line 134-136: sentence rephrased as suggested.

Line 263 etc.: “vacuole” changed to “vacuoles”

Line 470-472: reference to the Ciona case has been added. An example of beta-catenin's role in amphioxus is given. This paragraph has been re-edit to clarify the statement.

Figure 3 and paragraph on GRN 

I do not find the figure much improved. The discussion on the GRN remains confusing as the figure, at least the author should rearrange the figure in such a way that a clear distinction between patterning factors and differentiating TF factors would emerge, possibly within a GRN between the genes wherever is known at least for vertebrates and others.. along the lines of what published by Di Gregorio, 2020 https://doi.org/10.1016/bs.ctdb.2020.01.002.

Despite my comment, in the revised figures the authors failed to mention both Col2A1 in zebrafish notochord or Laminin, which is present in Zebrafish, Ciona and annelids. This should be mentioned.

Response:

We choose to focus on the three TF mentioned in the text in the discussion on GRN. While we are aware that these might not be the clearest choices, given the lack of information on the GRN in basal species, we hope this provides a framework for general notochord GRN and hope that future studies will elucidate the interactions between them. The purpose of showing figure 3 is mainly to signify the layered and divergent genetic regulations during notochord development exhibited in those related species. While we agree that more detailed GRN in ascidian and vertebrate could be provided based on existing data, we think introducing it there does not necessarily benefit understanding. Alternatively, we included references in the text to audience for the detailed previously published summary(Nori Satoh and Anna, Di Gregorio’s papers)“More detailed GRNs could be found in previously published reviews [77],[80]”.

Thank you again for your suggestions

Round 3

Reviewer 3 Report

The manuscript has been improved following most of the suggestions. Despite providing an account of the different hyopotheses and experimental work available on notochord evolution is always useful, in the revised form the work still lacks in novelty as compared to previous work addressing the issue. The GRN paragraph is improved but should be better structured not just be a bunch of sentences, the message should be clear, for instance it seems that b-catenin should be a paragraph on its own with a proper introductory sentence or should be embedded into the different discussion points (FoxA etc). 

Author Response

Thanks for the comments and suggestions. We have revised the first part of GRN part and provided a paragraph to clearly introduce beta-catenin. Please have a check.